# Age and Sexual Maturity Estimation of Stranded Striped Dolphins, *Stenella coeruleoalba*, Infected with *Brucella ceti*

**Karol Roca-Monge** [1], **Rocío González-Barrientos** [2], **Marcela Suárez-Esquivel** [1], **José David Palacios-Alfaro** [3], **Laura Castro-Ramírez** [1], **Mauricio Jiménez-Soto** [1], **Minor Cordero-Chavarría** [4], **Daniel García-Párraga** [5], **Ashley Barratclough** [6], **Edgardo Moreno** [1] and **Gabriela Hernández-Mora** [4],*

1   Escuela de Medicina Veterinaria, Universidad Nacional, Heredia 40104, Costa Rica
2   Texas A&M Veterinary Medical Diagnostic Laboratory, PO Drawer 3040, College Station, TX 77841-3040, USA
3   Independent Researcher, San Pablo de Heredia, Santiago, Heredia 40503, Costa Rica
4   Servicio Nacional de Salud Animal (SENASA), Ministerio de Agricultura y Ganadería, Heredia 40104, Costa Rica
5   Research Department, Fundación Oceanogràfic de la Comunitat Valenciana, Oceanogràfic, Ciudad de las Artes y las Ciencias, 46013 Valencia, Spain
6   National Marine Mammal Foundation, San Diego, CA 92107, USA
*   Correspondence: gabbytica@gmail.com or gabriela.hernandez.m@senasa.go.cr; Tel.: +506-25871837

**Abstract:** Age parameters in cetaceans allow examining conservation and studying individuals with growth affection. The age and sexual maturity of 51 stranded *Stenella coeruleoalba* striped dolphins from the Eastern Tropical Pacific (ETP) of Costa Rica, most suffering brucellosis (95.6%), were assessed. In order to ascertain the dolphins' ages, we measured the length and growth of dentin-layer group counts (GLGs) and assessed flipper bone radiography without (FBSA) and with a formula (FBF). Sexual maturity was determined through gonadal histology and sexual hormone serum levels. Compared with a model based on *S. coeruleoalba* ages estimations in other latitudes, the striped dolphin studied displayed deficient growth parameters, with considerable variability in length, teeth, and flippers bone development. Close to 43% (*n* = 15) of GLGs' measurements were below the body length average ranges for the predicted age, suggesting developmental abnormalities. Likewise, 34.4% and 31.2% of the dolphins assessed by FBSA and FBF were also below the body length based on age prediction curves, also indicating developmental abnormalities. This information is supported by the poor correlation between GLGs, FBSA, and FBF. Inconsistencies between sexually mature males and females related to GLGs, FBSA, and FBF were evident. Although the different oceanic settings of the ETP, such as contamination, food access, diseases, and other parameters, may influence size variation, our data also suggest that long-lasting debilitating brucellosis may account for detrimental growth in the ETP striped dolphins. Our study highlights the possible deleterious consequences of chronic infectious diseases in the cetacean populations already confronting distressful conditions.

**Keywords:** *Stenella coeruleoalba*; *Brucella ceti*; radiography; gonadal histology; GLG; sexual hormones; age estimation

## 1. Introduction

The striped dolphins (*S. coeruleoalba*) are among the most abundant cetaceans worldwide in tropical, subtropical, and temperate waters [1–4]. The abundance in the Eastern Tropical Pacific (ETP) region is estimated at 1.5 million and is categorized as of "Least Concern" [5]. This cetacean is in the Convention on Migratory Species, as with unfavorable conservation status, and requires international agreements for their protection, conservation, and management. This free-living dolphin species has never been successfully kept in captivity or rehabilitation for extended periods [2].

Most age, reproductive, and size parameters of *S. coeruleoalba* have been obtained through information from driven fisheries, bycatch, or stranded animals [2]. On average,

Mediterranean striped dolphins are 5–8 cm shorter than their Atlantic and Pacific counterparts [2]. The Western Pacific Ocean (WPO) *S. coeruleoalba* dolphins' mean body length is estimated at 236 cm for males and 220 cm for females [6]. However, based on the skull size, the WPO dolphins are, on average, larger than the ETP striped dolphins [1].

It has been shown that striped dolphins in the WPO rapidly increase in body length after birth (average of 100 cm), reaching lengths close to ~188 cm after two years [6], with sexual dimorphism beginning on average at 2.5 years after birth, with males exceeding the females in ~4 cm [6]. Sizes close to 175 cm commonly belong to weaning individuals. If females exceed 209 cm (at 5–13 years) and males 219 cm (at 7–15 years), they can be assumed to be mature adults [6]. After 8 years of age, the mean growth of the female becomes slower; this is why males attain the asymptotic length of 236 cm at the age of ~21 years, and the females reach the asymptotic length of ~225 cm at 17 years of age [6]. As in other mammals, under stressful conditions and reduced population density, sexual maturity in dolphins may be achieved earlier and shortened by two to three years of age [7].

Cetacean brucellosis due to *Brucella ceti* is a long-lasting debilitating disease that induces abortion and death of dolphins [8–13]. We and others have shown that this disease causes lesions in organs and bones, which hamper the dolphins' physiological performance and development. Suppurative, necrotizing, granulomatous epididymitis, orchitis, and azoospermia have been described in *Brucella*-infected male cetaceans. In *Brucella*-infected females, abortions, mastitis, abscesses, heart disease, caseous necrosis, calcified foci in the mammary gland, granulomatous and necrosuppurative endometritis, and placentitis have been reported in various latitudes [8–22]. In the final stages of the disease, the bacterium may cross the blood-brain barrier, causing meningoencephalomyelitis, swimming impairment, and death [8,9]. The prevalence of *Brucella* infections in cetaceans is high in the ETP [8,10], and strandings due to brucellosis are common.

In Costa Rica, striped dolphins correspond to 72% of the stranded in the ETP coast between 2001 and 2021. However, strandings of these animals also occur in countries alongside the Central American Pacific coast [9,23–25]. The primary cause of most striped-dolphin strandings in Costa Rica is neurobrucellosis, with close to 95% of seropositive animals and 77.5% successful isolation of *B. ceti*. This neurological infection impairs the survival of these animals because of swimming and buoyancy problems and hunting and feeding difficulties demonstrated by the absence of stomach contents in all these stranded animals [8,9,11,13,26]. This phenomenon can be extended to striped-dolphin strandings in El Salvador, from which we have also detected *B. ceti* infections, and striped dolphins from the Mediterranean Sea [27,28] and other latitudes [16,29]. However, the diagnosis of brucellosis in most areas where these dolphins strand is not routinely performed [30–32], even though it is known that neurological symptoms are frequently associated with brucellosis [8–13,27–29]. Therefore, the incidence of this infection in *S. coeruleoalba* is currently unknown but suspected to be high as stated 10 years ago [10] because, although it is more commonly detected [8–13,27–29], there are no systematic studies demonstrating the prevalence and incidence of the infection in the oceans

When dealing with the wild fauna, age estimation is essential to understand the reproductive status and the interplay that long-lasting debilitating diseases may have in sexual maturation within the framework of population dynamics [33–36]. However, in cetaceans, age estimation is complex as they do not show obvious external signs indicating their age. Therefore, various techniques have been developed. The most straightforward method is the analysis of total body length [4,6,35]. Other methods include counting dentin's growth layer groups (GLG) and bone development [34]. Age evaluation by estimating the bones' maturation of pectoral flippers has been described for various cetacean species, including *S. coeruleoalba* [7], and standardized for bottlenose dolphins (*Tursiops truncatus*) [33,36]. For sexual maturity estimation, the most reliable method has been a histological examination of the gonads and the measurement of steroid sexual hormones [37,38].

The stranded dolphins infected with *B. ceti* identified in Costa Rica represent an opportunity to evaluate the impact that a long-lasting debilitating disease, such as brucellosis,

may have on the physiological development of cetaceans and also to make an overview of the striped dolphins stranded in Costa Rica. Since we do not have a "normal" population of striped dolphins that could serve as a reference group for establishing specific age limits, we developed an alternative model based on growth curves for *Stenella coeruleoalba and Stenella* dolphins from different latitudes and discuss our results under this framework.

## 2. Materials and Methods

### 2.1. Dolphin Population

We used 51 striped dolphins, 28 males and 23 females, stranded on the ETP coast of Costa Rica from 2006–2021. All animals stranded alive and died within a few hours after the first report and, therefore, were code 2 of freshness [39]. Serology was performed as described before [11]. Necropsies were done following the USDA National Veterinary Services Laboratory guidelines [40] and SENASA rules of de Ministry of Agriculture of Costa Rica (Ley General del Servicio Nacional de Salud Animal N° 8495). Sex and body size were assessed and measured upon arrival, with a length defined as the tip of the rostrum to the caudal fluke notch. (Supplementary Data S1). Gonads were not measured or weighed. Neurobrucellosis infection in the studied dolphins has been thoroughly reported [8–11,13,26]. All affected dolphins showed brain abnormalities and swimming difficulties [8,9,11,13,26]. Classification of age was based on complementing three age methods.

### 2.2. Growth Layer Groups (GLG) of Dentin

Tooth age estimation was based on dentine because the cementum is relatively thin in homodonts and stops its accumulation early in life [41,42]. Four to six teeth were collected from the middle part of the jaw and stored at −80 °C. The teeth were cleaned, and gingival tissue was removed [43]. The teeth were immersed in blocks in self-curing acrylic, keeping them fixed for wear processing. A low-speed saw was used to make each tooth thinner on both sides until it was 1 mm thick. Teeth were stepwise polished using sandpaper of different grain sizes (1000–2000 lines/inch). Subsequently, they were left immersed in Leica® Harris Hematoxylin solution for 72 h. The stained teeth were washed and examined under a dark field microscope. Depending on the outcome, some teeth required exposure to formic acid for 30–60 s, followed by washing with water. In addition, conventional histology with decalcification and Mayer's hematoxylin stain was performed to confirm the GLGs in some animals [44]. The microscopic images were analyzed with Adobe Photoshop ®, and the GLGs were counted (Supplementary Data S1).

### 2.3. Bone Ossification

Pectoral flippers were subjected to X-ray radiography and analyzed as described [7,36], classifying the epiphyses in 6 stages based on their development, the description of which is: stage 0 = no secondary ossification center present; stage 1 = secondary ossification center is present but occupies less than 50% of the width of the adjacent bone; stage 2 = the secondary ossification center occupies 50–100% of the adjacent bone; stage 3 = the distance between the epiphysis and the metaphysis of bone begins to diminish; stage 4 = epiphyseal plate begins to close; stage 5 = physis is entirely closed, and a radio dense physeal line traverses the width of the bone; stage 6 = remodeling of the physical region, consisting of replacement of the physeal line by mature bone tissue, as a consequence, less than 50% to no evidence of the physeal line remains. Only radius, ulna, and metacarpals were estimated because the high individual variation of the phalanges in striped dolphins can lead to misinterpretation [7,45]. The age estimation was based on two parameters: estimation through semiquantitative evaluation of flipper bone stages analysis comparing with a catalog (FBSA) [7], and determination based on most indicative bones following the flippers bone formula (FBF) $x = 0.000383y^3 − 0.0019398y^2 + 0.1101071y − 0.2517343$ [33]. This last method includes delta bones and phalanges (as mentioned before, they are highly variable in *S. coeruleoalba*) (Supplementary Data S1). The formula was developed for

*T. truncatus*. Then, an intrinsic measurement error is expected when applied to *S. coeruleoalba*. However, since the formula is a dependent variable, the error was the same for all tested dolphins. We also determined degenerative, nutritional, or pathological changes in the fin bones that could conduct to misinterpretation of the ages.

## 2.4. Gonadal Histology

Gonads were fixed in 10% neutral buffered formalin, processed, and stained with hematoxylin and eosin [8,44]. Male gonads were explored for immature or mature germinal epithelium or mature spermatozoa [46]. The ovarian parenchyma was evaluated to identify developing cortical structures indicative of sexual maturity. The presence of corpora lutea, corpora albicans, or Graafian follicles was considered mature because the number of ovulations and *S. coeruleoalba* age does not come on a single linear relation. Moreover, this cetacean species only retains corpora albicans when they originated from the corpus luteum of pregnancy, and the corpus luteum degenerates ten days after ovulation [47,48]. In testes, the presence of spermatozoa was the principal indicator of maturity.

## 2.5. Hormonal Levels

Serum was collected at the time of stranding from live animals or cooled dolphins at 4 °C within 16 h postmortem. Sera were stored at −80 °C, and hormonal measurements were performed by automated competitive fluorescent enzyme-linked immunosorbent assay in an automated analyzer AIA-360® according to the manufacturer's instructions. For progesterone testing, a volume of 75 μL was used, and 85 μL for testosterone [49]. Less than 0.8 ng/mL of progesterone was estimated for pre-pubertal dolphins. According to *T. truncatus* parameters, progesterone levels >10 ng/mL may be related to pregnancy, while values higher than 1 ng/mL are associated with ovulation or regular post-pubertal numbers [50,51]. Females in anestrus or senescence present levels lower than 1 ng/mL [38]. In males, low testosterone levels do not necessarily equate to sexual immaturity, but higher levels are consistent with sexually mature or maturing cetaceans. Levels lower than 1 ng/dL suggest illness or stress factors [38] (Supplementary Data S1).

## 2.6. Statistical Analysis

Modeling of logarithmic growth curves (length versus age) followed three simulations according to growth data presented for *S. coeruleoalba* in the North and Central Pacific Ocean [1], Northwest Pacific Ocean in Japan [6], Southwest Mediterranean [7,52,53], and some delphinids from the Southwest Atlantic Ocean in Brazil [54]. The average logarithmic growth curve was between the upper and lower limits, within the range of *S. coeruleoalba* of the North and Central East Pacific Ocean. One-way analysis of variance (ANOVA) followed by Dunnett's test or multivariate analysis (MANOVA) was used for statistical significance. Data were processed in Microsoft Office Excel 365.

## 3. Results

Of the 51 animals in the study, 16 (31.4%) were adults, the majority were juveniles, with 28 individuals in this category (54.9%), 6 (11.8%) were pre-weaned, and 1 (1.9%) was a fetus. The dolphin sex ratio of the studied stranded population (male/female) was ~1.2. Brucellosis was confirmed by culture of *B. ceti* in 22 males (78.5%) and 18 females (78.2%). In addition, positive serology was described in 22 males (from the 24 sera) and 22 females (from the 22 sera). Due to the poor quality of some sera, we could not perform serology in four males and one female. Taking both the serological and bacteriological analysis, we diagnosed brucellosis in 22/28 males (78.5%) and 22/23 (96%), for a total of 86.2% positivity. However, the histopathological analysis of animals showing signs of meningoencephalomyelitis compatible with active brucellosis infections was ~ 98% (Supplementary Data S1).

As in other dolphins, GLGs contained high-density incremental lines according to developmental age, which could be counted, considering that each opaque and translucent

layer represents ~6 months according to the climate season [6] (Figure 1A). The correlation between length and GLGs was low, with no significant difference between males and females (Figure 1B). Of 35 dolphins with GLGs measurements, 15 (43%) were out of the body length range for the age prediction curves, suggesting developmental abnormalities (Figure 1C). From this, nine dolphins with abnormal growth (25.7%) presented lower body size in relation to the GLG growth, suggesting poor size development. Likewise, six (17.1%) dolphins with large body sizes presented few GLGs. Twenty dolphins (57%) were within the expected range of the predictive values (Figure 1C).

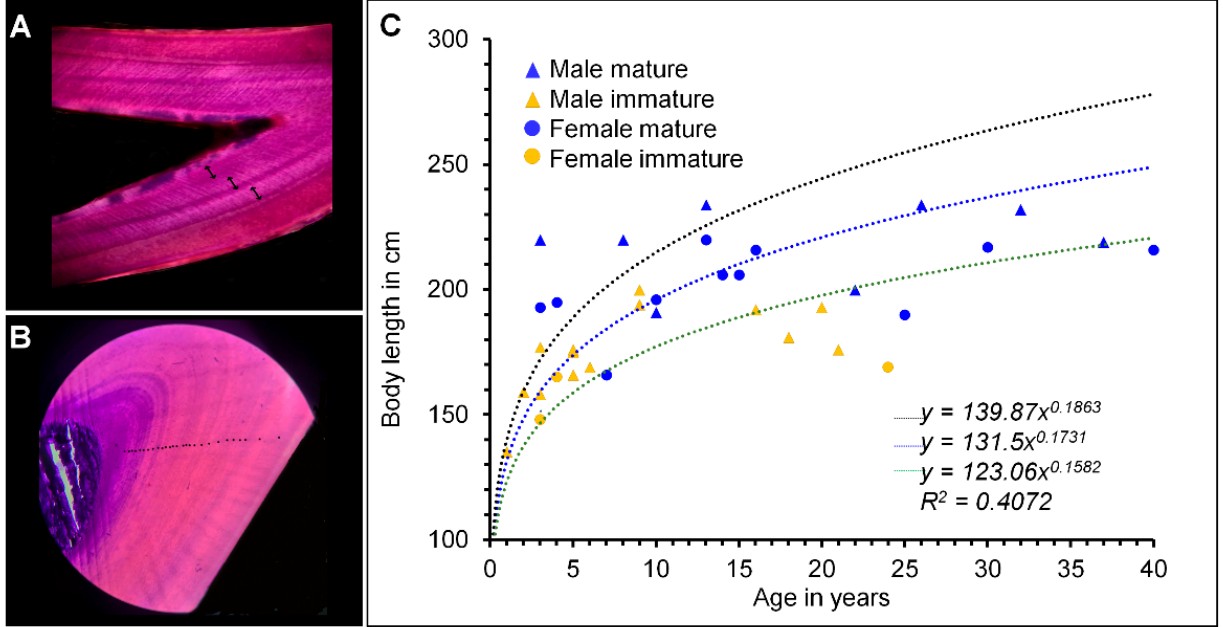

**Figure 1.** *S. coeruleoalba* age estimation according to body length and dentinal GLG groups. (**A**,**B**) Hematoxylin-stained teeth sections show the GLG groups (arrows and dots). (**A**) Female pre-weaned dolphin with 3 GLGs (CR07-19). (**B**) Female adult dolphin with 29 GLGs (CR03-20). (**C**) The dotted curves denote three dolphin age models according to *S. coeruleoalba* body length [6,7,52–54]. $R^2$ is the proportion of variation according to GLGs with body length. (Supplemental Data S1).

The coefficient of correlation between the FBSA related to the FBF corresponded to 0.86 ($R^2 = 0.81$) (Supplementary Data S1). Likewise, the correlations between the age estimation through FBSA and FBF and the expected dolphin lengths were 0.69 ($R^2 = 0.45$) and 0.69 ($R^2 = 0.52$), respectively (Figure 2B,C). However, while the FBSA age estimation displayed a scattered distribution along the prediction curves (Figure 2B), the FBF age estimation was arranged in two clusters: one <14 years old and one >28 years old (Figure 2C). The second older group differed from these dolphins' expected range of ages (42.4 ± 6.9 years old). Of 32 dolphins with FBSA and FBF measurements, 11 and 10 (34.4% and 31.2%, respectively) were out of the body-length age prediction curves, suggesting developmental abnormalities. Those below the bottom prediction curve displayed small size related to bone development, while those above the left of the upper prediction curve demonstrated poor bone maturation related to size. An alternative explanation, but unlikely, is that animals are larger for their age. However, all but one animal above the range were already sexually mature with high hormone levels (Supplementary Data S1).

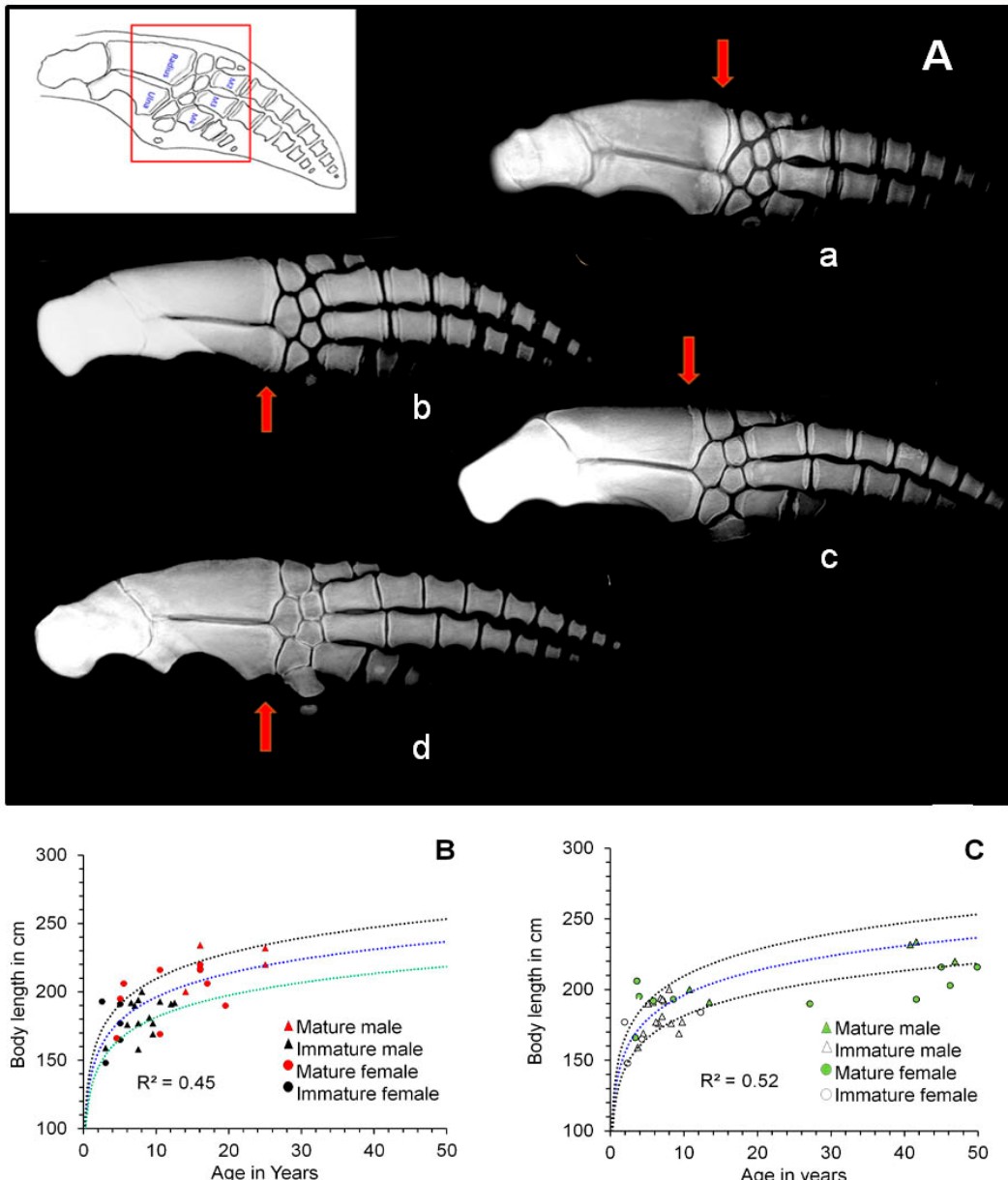

**Figure 2.** *S. coeruleoalba* age estimation according to body length and pectoral flippers bone characters. (**A**) X-ray radiography of the epiphysis of radius, ulna, and metacarpals of pectoral flippers was used for age evaluation (insert, red square insert). Arrows point to the radioulnar epiphysis used for age evaluation. (**a**) radio-ulna epiphyses radiolucent at stage 3 of pre-weaned female (CR05-20), of ~2.5 years old; (**b**) trabecular bridge visible of a juvenile female (CR01-18) epiphyses at stage 4 of ~ 7 years old; (**c**) ghost physis present in a female, young adult (CR05-19) at stage 5 of ~11.5 years old; and (**d**) adult male (CR10-20) at stage 6 of > 25 years old years, where remodeling of the physical region and replacement of the physeal line by mature bone tissue occurs. (**B**) Correlation between FBSA and body length. (**C**) Correlation between FBF (x = 0.000383y$^3$ − 0.0019398y$^2$ + 0.1101071y − 0.2517343) and body length. The dotted curves denote three dolphin age models according to *Stenella* dolphins' body length [3,6,53,54]. R$^2$ denotes the proportion of variation of age estimation according to flippers' bone characters with body length. (Details in Supplemental Data S1). The formulas for the slope of the curves are depicted in Figure 1.

Sexual maturity in *S. coeruleoalba* males is reached between 8 and 10 years, while in females between 7 and 9 years [6]. The criteria for both sexes were based on gonad histology. Males who presented small seminiferous tubules lined by developing germinal

epithelium without spermatozoa were immature (Figure 3A,B), while those with spermatozoa were mature (Figure 3C). When female dolphins reach maturity, ovaries present diverse structures and follicles, in which the cortex is rich in primordial follicles, and no other structures are apparent (Figure 3). Regarding the females, 6/23 were immature, with ages lower than five years, and 17/23 were mature (5 ≥166 cm and <189 cm). The panorama in males was the opposite: 20/28 were immature, and 8/28 were mature (≥191 cm). Again, inconsistencies between sexually mature males and females related to GLGs, FBSA, and FBF were evident (Figures 1 and 2). Overall, dolphins of ≥190 cm were mature (17/18 for GLG, 12/14 for FBSA, and 13/14 for FBF), while smaller dolphins (<189 cm) were mainly sexually immature (13/16 for GLGs, 11/16 for FBSA, and 14/19 for FBF).

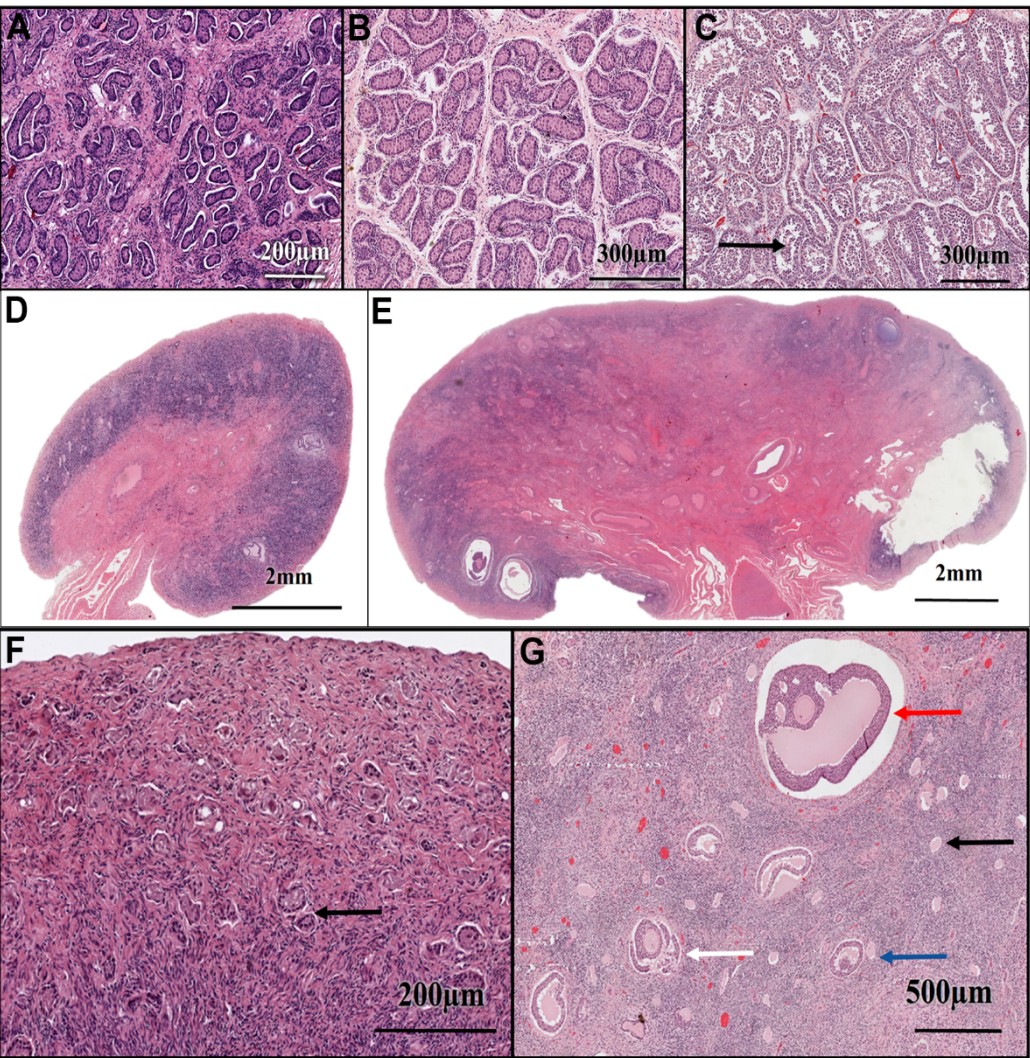

**Figure 3.** Gonads histology of *S. coeruleoalba* dolphins. (**A**) Testis with immature seminiferous tubules of a calf (CR22-21). (**B**) Immature seminiferous tubules of a juvenile male near puberty. No spermatozoa present (CR03-16). (**C**) Mature seminiferous tubules of an adult male (CR03-20). The black arrow points to spermatozoa within the lumen. (**D**) Right, and (**E**) left ovaries of a mature adult female *S. coeruleoalba* dolphin (CR21-21). The images show the asymmetry in size in a transversal view (5 mm versus 20 mm) and maturity between the right and left ovaries. (**F**) Histology of an immature ovary featuring numerous primordial follicles throughout the cortex (black arrow) (CR05-20). (**G**) Histology of a mature female shows follicles in different development stages (CR05-13). The black arrow depicts a primordial follicle, the blue arrow a primary follicle, the white arrow a secondary follicle, and the red arrow a Graafian follicle.

Figure 4 shows the distribution of hormones in male and female dolphins according to gonad sexual maturity. As expected, the difference in testosterone levels between dolphins with mature and immature testes was significant (<0.005). None of the males had testosterone levels lower than 1 ng/dL, indicating that none were in a state of senescence. On the other side, in females, the difference between the range of the immature (1.55 ± 1.45 ng/mL) and non-gravid mature (1.45 ± 0.69 ng/mL) dolphins was not significant. However, progesterone levels were considerably higher in the two pregnant female dolphins, which presented 18.31 ng/mL (CR08-06) and 12.58 ng/mL (CR04-12).

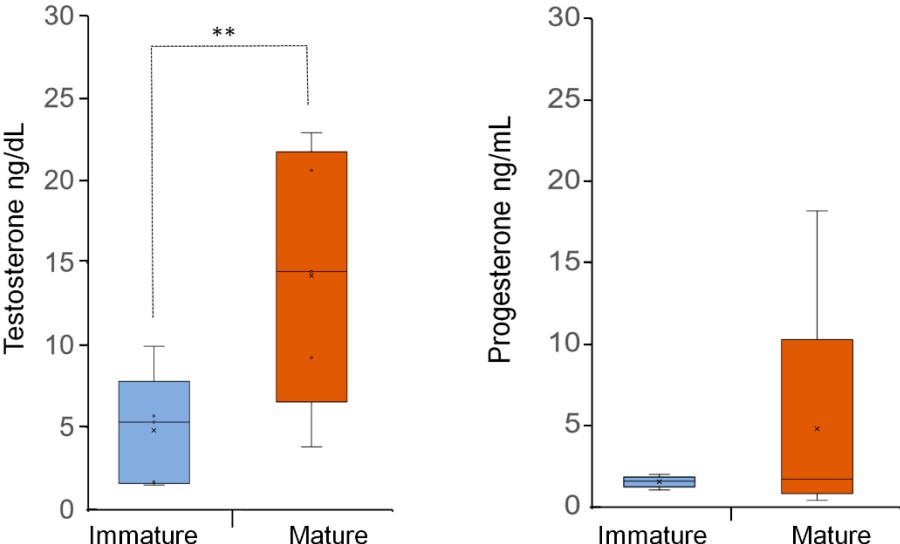

**Figure 4.** Distribution of mature and immature *S. coeruleoalba* striped dolphins according to sex hormones. Testosterone and progesterone levels were measured in blood obtained from dolphins at the time of stranding. Two female dolphins with progesterone levels >12 ng/mL were pregnant at the time of stranding (CR08-06 and CR04-12). Asterisk denotes a *p* < 0.005, top circle and bottom circle denotes quartile 3 and 1 respectively, X denotes mean.

## 4. Discussion

Striped dolphins are often divided into one of three social groups: juvenile, adult, and mixed; the last two are further divided into breeding and nonbreeding social groupings. Juvenile groups migrate closer to the coast than adults and mixed demographics. Therefore, young animals, mostly inshore, can increase the risk of stranding when sick or disorientated. In contrast, adult dolphins mainly inhabiting offshore waters would not reach the coast, and their corpses may be lost in the sea [35]. The pelagic range of the adult animals and lack of carcass recovery may cause a bias in our sample towards sexually immature animals, which are overrepresented. The sex ratio reported in the WPO population is 1.14 [6], similar to the 1.2 in this study. Previous studies in this oceanographic area have shown that the high ratio of stranded males at the age of 2 and 15 years in *S. coeruleoalba* is the result of the sexual segregation in the period between weaning and the attainment of full sexual maturity [55]. Calves remain in adult social groups until 1 or 2 years after weaning and then leave to join other juvenile dolphins. Males rejoin adult schools after reaching sexual maturity [35].

A significant number of dolphins did not match the expected length, teeth, and bone developments according to the proposed model based on striped-dolphin age parameters from various populations. These findings attract attention considering that the results of this study are from a stranded population with 95.6% (*n* = 46 available sera) seropositive animals for brucellosis and *B. ceti* was isolated from the central nervous system, specifically cerebrospinal fluid (CSF) in 78.4% (*n* = 40) of them. Meningoencephalomyelitis was also confirmed in 98% of them, and the severity of the inflammation causing secondary hydrocephalus has been graded using histopathology and axial computed tomography [9].

Although the bone formula for calculating age according to FBF parameters may introduce a measurement error when applied to *S. coeruleoalba,* the error should be the same for all analyzed dolphins. Still, the discrepancies between bone development and predicted size are significant. For instance, the discrepancies between GLGs and flipper bone assessments were significant (GLGs versus FBSA and FBF correlation were 0.69 [$R^2 = 0.67$] and 0.54 [$R^2 = 0.28$], respectively, (Supplemental Data S1)) and mostly did not coincide. It is worth emphasizing that, unlike the FBF, which was developed for *T. truncatus*, the methodology used in this investigation for skeletal maturation was developed to be applied to any cetacean [36]. Therefore, this scheme has been successfully used in many odontocetes species, including striped dolphins from the Mediterranean [7].

The ultrastructural variation and anomalies that GLGs can contain, such as marker lines, accessory lines, pulp stones, dental resorption, and cemental disturbance, can be a sign of environmental variations, such as El Niño and climate change and life history events. These events can interfere with the count of GLGs [42]. In addition, interference in the GLGs count due to tropical waters that do not have strictly delimited seasons should be considered.

Alternatively, part of the size variation of the ETP population related to striped dolphins from other oceanic latitudes could be due to different oceanic conditions. Indeed, the Pacific of Costa Rica is part of the tropical waters of the Costa Rican thermal dome. This zone is considered one of the most variable because of the influence of the Inter-Tropical Convergence Zone, in which winds generate seasonal changes and affect the oceanography of the area [56–58]. Moreover, these waters are a distinct biological habitat where phytoplankton and zooplankton biomass is higher than in surrounding tropical waters. The dome's physical structure and biological productivity affect the distribution and feeding of cetaceans, probably through forage availability. This oceanographic region constitutes the most critical area of overlap for striped dolphins, spotted dolphins (*Stenella attenuata*), spinner dolphins (*Stenella longirostris*), and common dolphins (*Delphinus delphis*) [56–58]; however, these last three species barely strand in the Costa Rican shores. In spinner dolphins, differences are found between dolphins from the Costa Rican area and other areas of the ETP; these fall into three geographical subspecies, with the "Central American type" (previously Costa Rican type) the longest [59–61]. Several other small odontocetes have also been found to vary geographically. Studies on variation in morphology indicate that population differentiation can occur in small cetaceans over relatively short distances [61].

In striped dolphins, the existence of two Eastern Pacific stocks, a Northern and a Southern one, was suspected, but no significant differences in body length were found years before between areas as measured by aerial photogrammetry [62]. However, sighting efforts had been relatively lower in the Central Tropical Pacific [63]. Consequently, there is still a possibility to consider a mild variation in the size of the striped dolphins present in Central American waters. Furthermore, it has been proposed that differences in thermoregulatory needs could account for a change in the length of animals as in terrestrial animals. If these populations experience higher temperatures on average than the rest of the species, then the ability to quickly shed excess metabolic heat via a reduction in body size would be selectively advantageous [1,64].

Although the different oceanic settings of the ETP, such as pollutants, food resources, other infectious diseases, and stress among several features, may be factors influencing size variation, our data indicate that a significant number of the dolphins studied (~45%) were below the expected length, bone, and teeth parameter proportions, indicating developmental deficiencies and pinpointing other causes. The fact that the dolphin population under study stranded alive and most of them were affected by active brucellosis, a long-lasting debilitating disease affecting the bones and several organs, suggests that at least part of the developmental abnormalities is because we were working with an ill population different from those studied after bycatch fisheries. It is worth noting that other diseases such as morbillivirus and herpes virus, or parasites such as toxoplasmosis, have been discarded

in these studied striped dolphins by the histology presentation, serology, and the use of next-generation sequencing of the CSF [8,9].

Brucellosis hampers reproduction in females and males, affecting the secondary reproductive organs in cetaceans, including striped dolphins [10]. As in cows, calves born from *Brucella*-infected mothers may be premature, weak, and unhealthy [65–67], affecting their performance in a highly competitive environment such as the open ocean. Indeed, several of these estimated juvenile striped dolphins should have shown signs of gonadal maturity and larger sizes. Sexually mature male dolphins with larger bodies are required to efficiently attend to females in the mating season and defend themselves against interactions with other males [68,69], a fact that does not seem evident in the population we studied. Moreover, males become socially mature at about 13.5 years old. Social maturity is when males may gain access to receptive females and successfully fertilize them [35,55]. The mean age of the male at the achievement of sexual maturity is estimated to be above 8.7 years; it was also reported that it can be extended to 12 to 13 years in some minorities [6].

Based on the average size of the dolphin females, more pregnant or lactating animals were expected; however, the absence of pregnancies could be due to the *B. ceti*-related pathologies that have been observed in livestock affected with brucellosis [66]. Alternatively, adult mature females may stay offshore and are less likely to strand. Consequently, both causes are not mutually exclusive and may work together. In cetaceans, reproductive problems in pregnant individuals such as placentitis are often reported [10]. One of the two pregnant females in the study (CR08-06) presented severe placentitis with multiple necrotic foci and a dead fetus. The dolphin was estimated to have been in the seventh month of gestation due to the fetus length (66 cm) and the presence of whiskers on the rostrum [11]. Adaptations should also be considered. Juvenile and adult survival rates, the average age at attainment of sexual maturity, pregnancy rates, and juvenile growth rates likely respond to changes in population status [70]. These adaptations have been demonstrated before on striped-dolphin females when populations declined between 1974 and 1992 because of exploitation by fisheries [71].

Our study did not show significant changes in seasonality. Just two female dolphins were beyond the lowest limit of progesterone, and no males showed signs of senescence (Supplementary data set S1). Progesterone is a very stable hormone in many conditions; neither hemolysis nor defrost or long-time freeze affects sample levels in different animals [72]. However, in cows, the quantity decreases fast in complete blood after collection [73]. For this reason, all the blood samples used for this study were collected from alive or recently dead animals. Some dolphins presented high levels of sexual hormones even though they showed an immature result in the histology examination of gonads, especially those suspected to be near puberty. Low testosterone levels do not necessarily equate to sexual immaturity, for example, when considering variations with breeding season, whereas higher levels are consistent with sexually mature or maturing dolphins. In addition, the health status of the animal affects testosterone levels in male dolphins, as previously reported [38].

Ovaries in mature females had asymmetry; this phenomenon is typical in many cetacean species [74,75]. Before sexual maturity, there is not a remarkable macroscopic asymmetry. An increase in size in the left ovary occurs at the onset of sexual maturity, even though a real difference is present in the neonatal stage. In striped dolphins, 93% of corpora albicans accumulate in the left ovary. On average, ovarian maturity occurs from the left ovary first, and many years later, the right ovary could still be immature. Nevertheless, the right ovary is not immature for life. After the maturity of the right ovary, the comparable corpora lutea accumulation rate becomes more in the right ovary than in the left ovary [74].

The method that better suited the growth-curves model was the flippers bone assessment. Although the FBF showed a better correlation than the FBSA, the dispersion of the FBSA was significantly more evenly distributed than the FBF, making the former more reliable than the latter, which gave disproportionate ages (>42 years) in the oldest group of dolphins. The formula developed for *T. truncatus* includes phalanges. However, the

individual intraspecific variation of these bones in striped dolphins is high and does not relate to age [7,45], a fact that affects the age distribution in the FBF measurements.

We cannot ensure that brucellosis in striped dolphins was the primary cause of growth-age discrepancies or the consequence of the deficient health status of the animals, whose faulty immune system finally favored the disease development. Affection by toxics such as high PCB levels can depress reproductive rates in striped dolphins. This phenomenon, documented in a Mediterranean population, should be included as a potential disturbance element in these animals [76,77]. In addition, PCBs can impact adult survival because of the immunocompromised state that causes and can increase vulnerability to pathogens [78]. Moreover, environmental detriment was observed in the Deepwater Horizon oil spill in Gulf of Mexico waters in years where *Brucella* diagnosis in bottlenose dolphins stranded on South California coasts was high (2011–2012). This phenomenon returned to normal in 2014, when the brucellosis infection levels were near the lower end compared to other years [79–81]. Prey availability, exposure to contaminants and biotoxins, parasitism, disease, and other stressors may play a role in determining an individual's overall body condition, which in turn may influence reproductive success and growth rate [79].

As mentioned, all stranded dolphins from our study (*n* = 51) had empty stomachs without any sign of a recent meal before the stranding. This incident means that the animals could not access food properly as part of the infection with *B. ceti* due to the impairment of our population's neurological functions and symptoms [8,9]. We and others have found high antibodies against *Brucella* organisms in cetaceans indicating infection but not necessarily the development of active disease [8,82]. However, the dolphin infection more closely resembles human infection, which may become active with evident pathological signs and general weakness [8]. The high latency potential of *B. ceti* has been previously suspected, where 44% of the *Brucella* cases in *T. truncatus* were young animals with a high degree of brucellosis manifestations [79]. Likewise, a report on dolphins from Oregon and Washington coasts presented similar demography. Overall, 80% of the *Brucella*-positive individuals with neurobrucellosis signs were reported to be subadults, including a Pacific white-sided dolphin (*Lagenorhynchus obliquidens*), striped dolphins, and short-beaked common dolphins (*Delphinus delphis*) [83].

Although abnormal age structures have been observed in stranded *S. coeruleoalba* in the Mediterranean Sea and are suggested to be related to infectious diseases such as Morvillivirius, no diagnosis or correlation to a specific disease has been performed [53]. Following this, no other study considering the use of more than one aging method in animals with brucellosis has been carried out, nor were sexual maturity studies of these animals performed simultaneously. Therefore, the correlation between different factors affecting these dolphins is, for the moment, not feasible. Our results are a first sight of the prevalence of abnormalities in a dolphin population with a chronic infectious disease such as brucellosis. The information presented here makes us aware of the deleterious consequences that chronic infectious diseases may have on cetaceans already confronting distressful conditions in the oceans.

**Supplementary Materials:** The following supporting information can be downloaded at: https://www.mdpi.com/article/10.3390/oceans3040033/s1, Supplemental Data S1.

**Author Contributions:** Conceptualization, R.G.-B. and G.H.-M.; methodology, R.G.-B, G.H.-M. and K.R.-M.; software, G.H.-M., K.R.-M. and E.M.; validation, R.G.-B., G.H.-M., K.R.-M. and E.M.; formal analysis, R.G.-B., G.H.-M., K.R.-M. and E.M.; investigation, K.R.-M., A.B., D.G.-P., R.G.-B. and G.H.-M.; resources, K.R.-M., M.S.-E., L.C.-R., M.J.-S., M.C.-C., R.G.-B. and G.H.-M.; data curation, K.R.-M., R.G.-B., M.S.-E., E.M. and G.H.-M.; writing—original draft preparation, K.R.-M., E.M. and G.H.-M.; writing—review and editing, K.R.-M., R.G.-B., M.S.-E., J.D.P.-A., L.C.-R., M.J.-S., M.C.-C., D.G.-P., A.B., E.M. and G.H.-M.; visualization, R.G.-B. and G.H.-M.; supervision, R.G.-B. and G.H.-M.; project administration, R.G.-B. and G.H.-M.; funding acquisition, K.R.-M., M.S.-E., L.C.-R., M.J.-S., M.C.-C. and G.H.-M. All authors have read and agreed to the published version of the manuscript.

**Funding:** This research was funded by FOCAES grant UNA-VI-OFIC-105-2021 for K.R.-M. and the National Wildlife Program of the National Animal Health Service (SENASA). Edgardo Moreno received support from the National Academy of Sciences of Costa Rica.

**Institutional Review Board Statement:** The study was conducted following the Declaration of Helsinki, approved by the Institutional Review Board of the National Brucellosis Control Program and Wildlife Program of the Costa Rican National Animal Health Service (SENASA), and performed in agreement with the corresponding law "Ley de Bienestar de los Animales"(Ley N_ 7451 1994) and to the International Convention for the Protection of Animals endorsed by Costa Rican Veterinary General Law on the SENASA (Ley N_. 8495 2006). All procedures involving live *Brucella* followed the "Reglamento de Bioseguridad de la CCSS 39,975-0, 2012" after the "Decreto Ejecutivo No. 30,965-S", the year 2002 and research protocol 0045-17 approved by the National University, Costa Rica. According to the Biodiversity Law No. 7788 of Costa Rica and the Convention on Biological Diversity, the genetic resources were accessed under the terms regarding an equal and fair distribution of benefits for those who provided resources under CONAGEBIO, Costa Rica, permit No. R-CM-UNA-003-2019-OT-CONAGEBIO.

**Acknowledgments:** We thank Jimmy Vargas Olivares from the Hospital San Vicente de Paul for assisting in the histopathological preparations, Andres Granados for his help in the necropsies, Karen Vega Benavides for X-ray radiographic images, and all the personnel of the National University Hospital for small species and wild animals. In addition, we thank Wayne McFee for consultation on tooth GLG aging and the personnel of SENASA, especially the Central Pacific Region and LANASEVE. We thank also the staff of the Parque Marino del Pacífico for their collaboration with the localization and handling of the stranded animals. We also thank the Municipal Police, the Coastguard of Costa Rica, and the Tropical Disease Investigation Program (PIET) personnel. This work was supported and approved as part of the National Program of Wildlife of SENASA San José, Costa Rica.

**Conflicts of Interest:** The authors declare no conflict of interest. Funders had no role in collecting, analyzing, interpreting data, writing the manuscript or publishing the results.

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
