# Peer review of "Age and Sexual Maturity Estimation of Stranded Striped Dolphins, Stenella coeruleoalba, Infected with Brucella ceti"

_2673-1924, doi:10.3390/oceans3040033_

Round 1

Reviewer 1 Report

This study reports the estimated age and sexual maturity of the beached striped dolphin, Stenella coeruleoalba, infected with Brucella ceti. In general, the article is original and considers a good number of samples since they are stranded animals. The results of the study are clearly presented and discussed, however there are some inaccuracies in the results and in the bibliography of the introduction. Here are some comments and suggestions that could improve the manuscript.

In the introduction “However, the diagnoses of brucellosis in most of these areas where these dolphins strand is not routinely performed [30- 32]”

In this sentence, I believe that the reference bibliography is not very significant because it refers only to the herpesvirus infection in the central nervous system of odontocetes or to the haematological and hematochemical values for the striped dolphin, without considering the diagnosis of brucella.

Still in the introduction “Therefore, the incidence of this infection in S. coeruleoalba is currently unknown but suspected to be high [10]”.

Unknown, I wouldn't say, your bibliographic reference is from 2012 there are many recent studies on brucellosis in Stenella coeruloalba.

In the results: “Brucellosis was confirmed by culture of B. ceti in 22 males (78.5%) and 18 females (78.2%).  Also, positive serology was described in 22 males (91.6%) (from the 24 available sera) and 22 females (100%) (from the 22 available sera)”. The total of animals in this study are 23 females and 28 males (51) with a ratio of 1.2. From what was written in the results, the serological positivity to Brucella in 46 total sera was confirmed only in 44 animals of which 40 have localization in the central nervous system. The 22 available sera of the females were all positive while for the males only 22, because 2 were negative. If I am not mistaken, in the supplementary data the serological positivity of Brucella males is 22 of the 25 available sera, of which 3 are negative, while in 3 indeterminate animals there is only 1 ND for females. In that case, you need to change the data. The question is why were the serum not collected from the 51 animals? In this work 4 animals were not taken into consideration, was the serum also taken? If the answer is no, the authors should indicate this more clearly in the materials and methods, because from reading it seems to have been collected in all animals. look down: “2.5. Hormonal levels. Serum was collected at the time of stranding from live animals or cooled dolphins at 4° C within 16 hours post-mortem”. 

In this sentence: “Of 35 dolphins with GLGs measurements, 15 (42.9%) were out of the body length range for the age prediction curves, suggesting developmental abnormalities (Figure 1B). Nine (25.7%) dolphins presented lower body size in relation to the GLG growth, suggesting poor size development. Likewise, six (17.1%) dolphins with large body sizes presented few GLG”.

This type of survey was only done on 35 dolphins. The text speaks of 15 out of the body length range, 9 with smaller body size and 6 with large body size. In the text we speak only of 30 animals, 5 are missing.

Sexual maturity in S. coeruleoalba is reached chiefly between 8 and 10 years in males, while in females between 7 and 9 years [6]. Since measurements of the gonads were not available, the criteria for both sexes were based on histology.

In the materials and methods, it is written “Gonads were collected and fixed in 10% neutral buffered formalin”. Why was it not possible to make this measurement?

“Some females”….. The sentence is interrupted then follows with “Figure 4 shows the distribution of hormones in male and female dolphins according to gonad sexual maturity”.

In the discussion: “The sex ratio reported in the WPO population is 1,14 [6], similar to the 1,2 in this study”.

Is WPO the Western Pacific Ocean? The abbreviation is not defined.

Check the bibliography because in some of them Brucella ceti are not written in italics.

Author Response

Answers to Reviewer 1:

This study reports the estimated age and sexual maturity of the beached striped dolphin, Stenella coeruleoalba, infected with Brucella ceti. In general, the article is original and considers a good number of samples since they are stranded animals. The results of the study are clearly presented and discussed, however there are some inaccuracies in the results and in the bibliography of the introduction. Here are some comments and suggestions that could improve the manuscript. 

We thank the reviewer for his/her comment

In the introduction "However, the diagnoses of brucellosis in most of these areas where these dolphins strand is not routinely performed [30- 32]" In this sentence, I believe that the reference bibliography is not very significant because it refers only to the herpesvirus infection in the central nervous system of odontocetes or to the haematological and hematochemical values for the striped dolphin, without considering the diagnosis of Brucella.

The reviewer's comment is well taken. However, our sentence's direction is precise, and the references are pertinent. Despite the dolphins in these works presenting neurological problems, they did not look for Brucella just for the virus. That is precisely our point. It is well known that clinical neurological symptoms are frequently associated with brucellosis in these animals. This event is relevant since coinfections of Brucella with the virus have also been observed. For clarity, we have rephrased the sentence (line 88-91).

Still in the introduction "Therefore, the incidence of this infection in S. coeruleoalba is currently unknown but suspected to be high [10]".

Unknown, I wouldn't say, your bibliographic reference is from 2012 there are many recent studies on Brucellosis in Stenella coeruloalba.

Following the reviewer's concern, we have included several newer references in the sentence and rephrased it to make it more straightforward. Despite this, the value of this reference stands even though it is ten years old. As of ten years ago, the incidence and prevalence of brucellosis in cetaceans remain unknown. There is not (yet) a single systematic epidemiological study showing the incidence and prevalence of brucellosis in the oceans and any cetacean species. Because strandings due to brucellosis are diagnosed more often nowadays, we suspect that its incidence is high (as stated in this reference, which caught this fact's attention for the first time), but we do not know for sure. (line 91-94).

In the results: Brucellosis was confirmed by culture of B. ceti in 22 males (78.5%) and 18 females (78.2%).  Also, positive serology was described in 22 males (91.6%) (from the 24 available sera) and 22 females (100%) (from the 22 available sera)". The total of animals in this study are 23 females and 28 males (51) with a ratio of 1.2. From what was written in the results, the serological positivity to Brucella in 46 total sera was confirmed only in 44 animals of which 40 have localization in the central nervous system. The 22 available sera of the females were all positive while for the males only 22, because 2 were negative. If I am not mistaken, in the supplementary data the serological positivity of Brucella males is 22 of  25 available sera, of which 3 are negative, while in 3 indeterminate animals there is only 1 ND for females. In that case, you need to change the data. The question is why were the serum not collected from the 51 animals? In this work 4 animals were not taken into consideration, was the serum also taken? If the answer is no, the authors should indicate this more clearly in the materials and methods, because from reading it seems to have been collected in all animals. look down: "2.5. Hormonal levels. Serum was collected at the time of stranding from live animals or cooled dolphins at 4° C within 16 hours post-mortem". 

We thank the reviewer for his/her criticisms and for calling our attention to this subject. We have thoroughly corrected the numbers and added a new column in the Supplementary data set indicating the presence of meningonecephalomylitis as an indication of presumptive (clinical brucellosis). Indeed we collected sera in all animals at the time of stranding. However, the quality of some sera was not good in a few cases due to technical problems or the animal's poor condition. We have clarified this in the corresponding sections (Lines,199-206).

In this sentence: "Of 35 dolphins with GLGs measurements, 15 (42.9%) were out of the body length range for the age prediction curves, suggesting developmental abnormalities (Figure 1B). Nine (25.7%) dolphins presented lower body size in relation to the GLG growth, suggesting poor size development. Likewise, six (17.1%) dolphins with large body sizes presented few GLG".

This type of survey was only done on 35 dolphins. The text speaks of 15 out of the body length range, 9 with smaller body size and 6 with large body size. In the text we speak only of 30 animals, 5 are missing.

The reviewer's comment is well taken. Although the numbers are correct, we agree that the sentence is somewhat confusing, so we have rephrased this paragraph accordingly (Lines 212-217)

Sexual maturity in S. coeruleoalba is reached chiefly between 8 and 10 years in males, while in females between 7 and 9 years [6]. Since measurements of the gonads were not available, the criteria for both sexes were based on histology.

In the materials and methods, it is written "Gonads were collected and fixed in 10% neutral buffered formalin". Why was it not possible to make this measurement?

Following the reviewer's comment, we have removed this sentence from the text to make it unambiguous (Line 165-166) In our experience, measuring and weighing the gonads of dolphins, particularly in juveniles, is a gross method and, in many cases, useless since there is much variation in dolphins. However, we understand that some researchers use it as a parameter to determine maturity. In this direction, we prefer a more accurate and objective measurement, such as histology and hormone detection, to determine maturity. We have removed this sentence from the text to make it unambiguous.

"Some females"….. The sentence is interrupted then follows with "Figure 4 shows the distribution of hormones in male and female dolphins according to gonad sexual maturity".

We thank the reviewer. The sentence was corrected accordingly

In the discussion: "The sex ratio reported in the WPO population is 1,14 [6], similar to the 1,2 in this study".

Is WPO the Western Pacific Ocean? The abbreviation is not defined.

We thank the reviewer. The abbreviation was removed and spelled out in full text line 55

Check the bibliography because in some of them, Brucella ceti are not written in italics.

All the Latin names have been checked and changed to italics.

Reviewer 2 Report

This is a well-organized and well written manuscript that presents compelling, solid evidence for the impact of brucellosis on stiped dolphins in the ETP. The ms seems to be in fine shape. The figures are useful, and the cited references are thorough and up to date.

I found very few things to be fixed in the writing, although line 260 seems incomplete, with an unfinished sentence starting with “Some females”…

Eastern Pacific should also always be capitalized (for example, in line 352, where it is “eastern pacific”).

As for the findings and conclusions, I am confident that the authors’ data and analysis are sound, and the Discussion presents a good perspective on this issue. The treatment of bone and tooth growth, how brucellosis likely affects reproduction, and why this study may be biased toward younger specimens are all handled well.

The authors do a good job explaining how conditions in the ETP could variously affect the health and ecology of Stenella in that region. The authors have also cited many examples of dolphins being infected with Brucella ceti in other ocean regions. Have the same deleterious consequences of brucellosis as described here for ETP striped dolphins been found in other regions and other species? What are some specific ways that the findings reported here differ from previous reports? This ms considers such information, but does not always present it in a way that a reader can quickly and clearly see what’s special about the stranded dolphins studied for this project.

Author Response

Answers to Reviewer 2:

This is a well-organized and well-written manuscript that presents compelling, solid evidence for the impact of brucellosis on striped dolphins in the ETP. The ms seems to be in fine shape. The figures are useful, and the cited references are thorough and up to date.

We thank the reviewer.

I found very few things to be fixed in the writing, although line 260 seems incomplete, with an unfinished sentence starting with "Some females"…

We thank the reviewer. The incomplete sentence was removed accordingly.

Eastern Pacific should also always be capitalized (for example, in line 352, where it is "eastern pacific").

We thank the reviewer. We have checked all the corresponding nouns and capitalize them accordingly.

As for the findings and conclusions, I am confident that the authors' data and analysis are sound, and the Discussion presents a good perspective on this issue. The treatment of bone and tooth growth, how brucellosis likely affects reproduction, and why this study may be biased toward younger specimens are all handled well.

We thank the reviewer's comment.

The authors do a good job explaining how conditions in the ETP could variously affect the health and ecology of Stenella in that region. The authors have also cited many examples of dolphins being infected with Brucella ceti in other ocean regions. Have the same deleterious consequences of brucellosis as described here for ETP striped dolphins been found in other regions and other species? What are some specific ways that the findings reported here differ from previous reports? This ms considers such information, but does not always present it in a way that a reader can quickly and clearly see what's special about the stranded dolphins studied for this project

The reviewers comment is pertinent. Accordingly, we have included several paragraphs of previous reports' findings in the discussion section. That may clarify better our contribution Line 427-431, 443-455

Round 2

Reviewer 1 Report

Dear authors,

You have accepted my suggestions and your explanations are clear.

Thanks

Author Response

We thank the reviewer for the comment

Reviewer 2 Report

I thank the authors for considering and following my suggestions.

Author Response

We thank the reviewer for the comment